# Expression and Prognostic Characteristics of Ferroptosis-Related Genes in Colon Cancer

**DOI:** 10.3390/ijms22115652

**Published:** 2021-05-26

**Authors:** Jie Zhu, Weikaixin Kong, Zhengwei Xie

**Affiliations:** 1Department of Pharmacology and International Cancer Institute, School of Basic Medical Sciences, Peking University, Beijing 100191, China; 1811210064@bjmu.edu.cn; 2Department of Molecular and Cellular Pharmacology, School of Pharmaceutical Sciences, Peking University Health Science Center, 38 Xueyuan Road, Haidian District, Beijing 100191, China; 1510307407@pku.edu.cn

**Keywords:** colon cancer, ferroptosis-related genes, prognosis

## Abstract

Ferroptosis is a new type of programmed cell death, which occurs with iron dependence. Previous studies have showed that ferroptosis plays an important regulatory role in the occurrence and development of tumors. Colon cancer is one of the major morbidities and causes of mortality in the world. This study used RNA-seq and colon cancer clinical data to explore the relationship between ferroptosis-related genes and colon cancer. Based on the fifteen prognostic ferroptosis-related genes, two molecular subgroups of colon cancer were identified. Surprisingly, we also found cluster2 was characterized by lower mutation burden and expression of checkpoint genes, better survival, and higher expression of *NOX1*. Moreover, cluster2 has fewer BRAF mutations. We also found the expression of *NOX1* is related to the status of BRAF. Finally, using 15 ferroptosis-related genes from The Cancer Genome Atlas cohort, we constructed a prognosis model, and this model may be used to predict the prognosis of patients in clinics.

## 1. Introduction

Colon cancer remains the second most common cause of cancer and cancer deaths worldwide. At least 25% of patients still present with metastatic disease, and at least 25–30% will develop metastatic colon cancer in the course of their disease [1,2]. According to the Cancer Statistics of 2020, approximately 140,000 new cases of colon cancer were diagnosed and 53,000 deaths from colon cancer were reported in the USA [3]. The incidence and mortality of colorectal cancer (CRC) have been increasing over the last 25 years in young adults below the age of 50. Gene mutation (K-ras, p53, B-raf proto-oncogene serine/threonine kinase (BRAF), mismatch repair genes) and microsatellite instability lead to the development of colon cancer [4]. Evidence suggests that modifiable lifestyle factors, including excess adiposity, poor diet, and physical inactivity, play an important role in the occurrence and progression of this disease [5,6]. To better the prognosis of patients with colon cancer, it is essential and urgent to identify new indicators for the prognosis evaluation and targeted therapy of colon cancer.

Ferroptosis is a new type of programmed cell death, which occurs with iron dependence [7]. Ferroptosis-inducing factors can directly or indirectly affect glutathione peroxidase through different pathways, resulting in a decrease in antioxidant capacity and accumulation of lipid reactive oxygen species (ROS) in cells, ultimately leading to oxidative cell death [8]. It is a complex and vulnerable process affected by a variety of crosslinks among mitochondrial, P53, iron metabolism, lipid metabolism, and oxidative stress [9,10,11]. Previous studies have shown that ferroptosis plays an important regulatory role in the occurrence and development of many diseases, such as tumors, neurological disease, acute kidney injury, ischemia/reperfusion, etc. [8]. Other studies also demonstrated that ferroptosis is an important mechanism by which cisplatin has been found to induce ferroptosis and apoptosis in CRC cells. Xie et al. [12] found that P53 inhibits erastin-induced ferroptosis in CRC cells by blocking the activity of dipeptidyl-peptidase-4 (DPP4), which was different from the previously discovered role of P53 in promoting ferroptosis in other cancer cells [13]. Immunotherapy represented by an immunological checkpoint blockade has demonstrated astounding clinical efficacy in a small percentage of patients with durable response [14]. It has been reported that CD8+T activation cells could enhance ferroptosis-specific lipid peroxidation in tumor cells, thus increasing ferroptosis and contributing to the anti-tumor efficacy of immunotherapy [15,16].

Based on the above findings, this study aimed to investigate the role of ferroptosis-related genes in colon cancer from the Gene Expression Omnibus (GEO) and The Cancer Genome Atlas (TCGA) cohort and evaluate the prognostic value for colon cancer patients of ferroptosis-related genes.

## 2. Results

### 2.1. The Landscape of Genetic Variation of Prognostic Ferroptosis-Related Genes in Colon Cancer

A total of 60 ferroptosis-related genes [17] were finally identified in this study. A univariate Cox regression model revealed 60 ferroptosis-related genes’ prognostic values in patients with colon cancer, in which fifteen genes (*ABCC1*, *ACACA*, *CISD1*, *CRYAB*, *CS*, *FANCD2*, *FDFT1*, *GLS2*, *GSS*, *HMGCR*, *KEAP1*, *LPCAT3*, *NFS1*, *NOX1*, *PEBP1*) were associated with prognosis (Figure 1A). These genes showed enrichment of biological processes related to the mitochondrion and metabolic process (Figure 1B). Figure 1C shows the mRNA expression levels of fifteen prognostic ferroptosis-related genes between normal and colon cancer samples in the GEO database. We found tumors showed a high expression of *ABCC1*, *ACACA*, *FANCD2*, *GLS2*, *GSS*, *KEAP1*, *NFS1*, and *PEBP1* and a low expression of *HMGCR* and *LPCAT3* compared to that of normal samples (Figure 1C, TCGA database was shown in Appendix A). It was found that tumors with a high expression of *FDFT1* showed a low expression of the *CRYAB* and a high expression of the *HMGCR*. Tumors with high expression of *GSS* showed a high expression of *NFS1*, *ABCC1*, and *NOX1* (Figure 1D). Among 399 samples, 69 experienced mutations of 15 prognostic ferroptosis-related genes, with a frequency of 17.29%. It was found that *ACACA* (32/69) exhibited the highest mutation frequency followed by *ABCC1* (23/69), while *CISD1* as well a *PEBP1* did not show any mutations in colon cancer samples (Figure 1E,F).

### 2.2. Construction of Two Molecular Subgroups of Colon Cancer Using Fifteen Ferroptosis-Related Genes with Prognoses

According to the expression of 15 ferroptosis-related genes associated with prognosis, patients were classified into qualitatively different molecular subgroups of colon cancer using the R software packages of ConsenesClusterPlus, and two molecular subgroups were eventually identified using unsupervised clustering, including 188 cases in cluster1, and 374 cases in cluster2 (GEO database shown in Figure 2A,B, TCGA shown in Appendix A). Comparison of the prognoses of the two molecular subgroups of colon cancer revealed a particularly prominent survival advantage in cluster2 (Figure 2C and Appendix A). A significant distinction existed in the transcriptional profile of the 15 ferroptosis-related genes between the two molecular subgroups of colon cancer. Cluster1 was characterized by the increased expression of *ACACA*, *CISD1*, *FANCD2*, and *GLS2* and presented a significant decrease in *NOX1*. We also noted that tumors with the cluster2 had a higher expression of *NOX1*, *NFS1*, and *GSS* (Figure 2D). Notably, both the TCGA and GEO databases showed that cluster2 with a high expression of *NOX1* had prominent survival (Figure 2D and Appendix A). To further investigate each molecular subgroup’s potential biological behavior, we determined 621 differentially expressed genes (DEGs) in two molecular colon cancer subgroups. The clusterProfilter package was used to perform GO enrichment analysis for DEGs. Surprisingly, the up DEGs in cluster2 were enriched in response to drugs, indicating the patients in cluster2 are sensitive to drugs (Figure 2E). The up DEGs in cluster1 presented enrichment in pathways associated with the inflammatory response, chemokine-mediated signaling pathways, and cell adhesion. Activation of these signaling pathways in cluster1 may be conducive to the metastasis of cancer (Figure 2F).

### 2.3. The Characteristics of Two Molecular Subgroups of Colon Cancer

Based on the GO enrichment analysis that cluster1 is related to both the immune response and inflammatory response, we then used a deconvolution algorithm based on support vector regression, CIBERSORT method, to determine the type of immune cells in the tumors, and we compared the component differences in immune cells between the two clusters. To our surprise, cluster1 had more M0/M1 macrophages, neutrophils, and gamma delta T cell. Compared with cluster1, cluster2 had more resting dendritic cells, plasma cells, and resting memory CD4 T cells (Figure 3A). As we know, the high expression of the immune checkpoint genes is another characteristic of immunosuppression. Thus, we examined the expression of these genes in the two clusters (Figure 3B). Consistent with the previous analysis, the expression of the immune checkpoint in cluster1 was higher than that of cluster2, indicating that patients in cluster1 are more likely to have an immune escape.

Then, we analyzed the distribution differences of somatic mutation between cluster1 and cluster2 in the TCGA-COAD cohort using the maftools package. The TMB quantification analyses confirmed cluster1 was markedly corelated with higher TMB (Figure 3C). We also found that cluster1 presented more extensive tumor mutation burden than did cluster2 (Figure 3D,E and Appendix A). Accumulated evidence demonstrated patients with high TMB status presented a durable clinical response to anti-PD-1/PDL1 immunotherapy [18]. Therefore, the above results indirectly demonstrated that the patients in cluster1 are more likely to receive immunotherapy.

### 2.4. The Role of Nox1 in Survival

To further investigate each cluster’s mutation genes, we determined 13 significantly differentially mutation genes. Notably, BRAF mutation is more frequent in cluster1 (Figure 4A). In metastatic colorectal cancer (mCRC), BRAF mutation represents a poor prognostic factor and median survival [19]. Combined with the previous results that *NOX1* had significantly lower expression in cluster1, we suspected that the poor survival of cluster1 with low expression of *NOX1* may be related to BRAF mutation. Next, we explored the relationship between *NOX1* genes and clinical factors. Surprisingly, the BRAF wild-type group had a high expression of *NOX1* (Figure 4B). We also observed that *NOX1* is related to the T/N stage and the state of MMR (Figure 4C–F). The above results showed that the patients in early colon cancer stages and fewer BRAF mutations had high expression of *NOX1*, consistent with previous results that the patients with higher *NOX1* expression had a better survival. Furthermore, we explored the genes related to *NOX1* in both the GEO39582 and TCGA databases. Then, we obtained nine genes (*NFE2L3*, *TSPAN6*, *PGRMC1*, *PLCB4*, *OCRL*, *CTPS2*, *XKRX*, *CFTR*, *ST6GAL1*, *NOX1*), among which *TSPAN6* and *PLCB4* are most relevant (Figure 4G–J and Appendix A). To further verify that *NOX1*-related genes are related to survival, based on the obtained 9 *NOX1*-related genes and *NOX1*, unsupervised clustering analyses were performed to further validate that these genes are related to survival. Finally, we found these genes can distinguish survival in both the GSE39582 and TCGA databases (Figure 4K,L). Therefore, *NOX1* and *COX2* (*PTGS2*) both play key roles in cancer development via the generation of an inflammatory microenvironment [20]. We next investigated the relationship between *NOX1*, *COX2*, and BRAF mutations. We found the expression of *NOX1* is negatively related to *COX2*. In the BRAF wild group, the expression of *NOX1* was high, while the expression of *COX2* was low (Appendix A). Previous studies showed that COX2 inhibitors, such as nonsteroidal anti-inflammatory drugs (NSAIDs), may decrease the risk and improve prognosis of carcinogenesis of various types of cancer including colorectal cancer [21,22,23,24]. The above results showed that *NOX1* genes played a non-negligible regulation role in survival.

### 2.5. Establishment of the Prognostic Model

Considering the importance of ferroptosis-related genes in cancer, we constructed a prognostic model based on fifteen ferroptosis-related genes. Three hundred and ninety-one patients were divided into training set (*n* = 196) and test set (*n* = 195). The clinical data and grouping of patients are shown in Table 1. To illustrate the random grouping’s rationality, differences in clinical variables between training set and test set were examined. Fifteen variables were used to establish a multivariable Cox regression model, and the independent variables were selected by back-off method. Finally, four variables were included in the Cox regression model (Figure 5C). The model contained genes related to ferroptosis, and we called this model the ferroptosis risk score. The formula of the ferroptosis risk score was as follows:Ferroptosis risk score = −0.08318 × *ISD1* − 0.01483 × *GSS* − 0.02273 × *FDFT1* + 0.00441 × *PEBP1*

The levels of the four genes are the normalized values of FPKM (fragments per kilobase of exon model per million mapped fragments).

According to the median value 1.12 of the ferroptosis risk score in the training set, the training and test sets were divided into high or low risk groups: patients with a ferroptosis risk score of less than 1.12 were in the low-risk group and those with a ferroptosis risk score greater than 1.12 were in the high-risk group. Moreover, Kaplan–Meier curves were used for survival analysis. We found that the high ferroptosis risk score had less survival time in both the training set (Figure 5A) and the test set (Figure 5B). The patients were classified according to the ferroptosis risk score, and the distribution of patient survival status was plotted. In both the training set and test set, patient survival time gradually decreased, and the number of deaths gradually increased with the increase in the ferroptosis risk score (Figure 5D,E, Appendix A). We used the ferroptosis risk score in the training set and test set to predict patient survival status at 1, 3, and 5 years, and plotted the receiver operator characteristic curves. The area under the curve (AUC) of the ferroptosis risk score in the training set for 1, 3, and 5 years was 0.672, 0.724, and 0.621, respectively (Appendix A). The AUC of the ferroptosis risk score in the test set for 1, 3, and 5 years was 0.645, 0.663, and 0.628, respectively (Appendix A). In the high ferroptosis risk score group, patients had lower expression of *FDFT1*, *GSS*, and *CISD1* and higher expression of *PEBP1* (Appendix A). The above results indicated that the prognostic model based on ferroptosis-related genes was effective.

### 2.6. Verification of Ferroptosis Risk Score

To further verify our prognostic model’ effectiveness, we applied the prognostic model established in the TCGA cohort to other independent colon cancer cohorts to verify its prognostic value. The ability of the ferroptosis risk score to predict relapse-free survival was evaluated in the GEO database (GSE39582). Consistent with the results in the TCGA cohort, patients with high ferroptosis risk scores had poorer survival than did the patients with low ferroptosis risk scores (Figure 6A). The area under the curve (AUC) of the ferroptosis risk score in the training set for 3 and 5 years was 0.609 and 0.582, respectively (Figure 6B,C). The survival time of patients gradually decreased, and the number of deaths gradually increased with the increase in the ferroptosis risk score (Figure 6D,E). In the high ferroptosis risk score group, patients had lower expression of *FDT1*, *GSS*, and *CISD1* and higher expression of *PEBP1* (Figure 6F). These findings provide evidence supporting that ferroptosis risk score is closely related to survival.

## 3. Discussion

Ferroptosis is a form of regulated cell death characterized by the iron-dependent accumulation of lipid hydroperoxides to lethal levels. Emerging evidence suggests that ferroptosis represents an ancient vulnerability caused by the incorporation of polyunsaturated fatty acids into cellular membranes, and that cells have developed complex systems that exploit and defend against this vulnerability in different contexts. Ferroptosis has been implicated in the pathological cell death associated with degenerative diseases (such as Alzheimer’s, Huntington’s, and Parkinson’s diseases), carcinogenesis, stroke, intracerebral hemorrhage, traumatic brain injury, ischemia-reperfusion injury, and kidney degeneration in mammals and is also implicated in heat stress in plants [25,26]. Ferroptosis may also have a tumor suppressor function that could be harnessed for cancer therapy [10].

Here, based on 15 prognostic ferroptosis-related genes, we revealed two molecular subgroups of colon cancer. Compared to cluster1, cluster2 was characterized by a lower expression of checkpoint genes, more DC cells and plasma cells, lower TMB, higher expression of *NOX1*, and lower BRAF mutants, corresponding to better survival. Cluster1 was characterized by activation of the inflammatory response; cell adhesion; cytokine–cytokine receptor interaction; more M0/M1 macrophages, neutrophils, and gamma delta T cells; and higher expression of checkpoint genes, indicating that patients in cluster1 are more likely to have an immune escape, and receive immunotherapy better.

Furthermore, in this study, we found that *NOX1* is significantly correlated with BRAF status. Firstly, cluster2, with higher expression of *NOX1*, had better survival and lower BRAF status, indicating the relationship among *NOX1*, BRAF, and survival. Therefore, based on the above results, we studied the genes related to *NOX1*, and found 9 genes (*NFE2L3*, *TSPAN6*, *PGRMC1*, *PLCB4*, *OCRL*, *CTPS2*, *XKRX*, *CFTR*, *ST6GAL1*, *NOX1*) that were positively associated with *NOX1*. Then, two subgroups of colon cancer were identified on the basis of the expression of these genes, which were also significantly correlated with survival. Moreover, we also found the *NOX1* was negatively related to *COX2*, and a *COX2* inhibitor is better for patients’ prognoses. Finally, we explored the feasibility of the 15 ferroptosis-related genes in colon cancer prognosis estimation and then constructed a prognostic model with these related genes. The prognostic model, ferroptosis risk score, built in this study provided a reference for treating patients with colon cancer.

In this model, high expression of *CISD1*, *GSS*, and *FDFT1* were favorable factors for prognosis. These genes could be roughly classified into iron metabolism (*CISD1*), lipid metabolism (*PEBP1*, *FDFT1*), and amino acid and glutathione metabolism (*GSS*) [10,27,28]. In terms of iron metabolism, *CISD1* encodes a protein with a CDGSH iron-sulfur domain and has been shown to bind a redox-active [2Fe-2S] cluster. The encoded protein has been localized to the outer membrane of mitochondria and is thought to play a role in limiting mitochondrial lipid peroxidation, negatively regulating ferroptosis cancer cell death [27,29]. The classical ferroptosis inducer erastin promoted *CISD1* expression in an iron-dependent manner in human hepatocellular carcinoma cells. Genetic inhibition of *CISD1* increased iron-mediated intramitochondrial lipid peroxidation, which contributes to erastin-induced ferroptosis [28]. Previous studies have shown that *CISD1* was related to survival of glioma [30], and high *CISD1* expression is a marker of better outcome in glioma [31]. Glutathione is important for a variety of biological functions, including protection of cells from oxidative damage by free radicals, detoxification of xenobiotics, and membrane transport. *GSS* encodes protein functions such as a homodimer to catalyze the second step of glutathione biosynthesis, which is the ATP-dependent conversion of gamma-L-glutamyl-L-cysteine to glutathione [32]. *FDFT1* encodes a membrane-associated enzyme located at a branch point in the mevalonate pathway, this enzyme is the first specific enzyme in cholesterol biosynthesis, catalyzing the dimerization of two molecules of farnesyl diphosphate in a two-step reaction to form squalene. Evidence suggests that fasting upregulates the expression of *FDFT1* during the inhibition of CRC cell aerobic glycolysis and proliferation. In addition, the downregulation of *FDFT1* is correlated with malignant progression and poor prognosis in CRC [33]. *PEBP1*, a scaffold protein inhibitor of protein kinase cascades, regulates ferroptotic cell death by binding with lipoxygenases and allowing them to generate lipid peroxides [34]. *PEBP1* affects various diseases including cancer, Alzheimer’s disease, and pancreatitis, which makes it a logical target for individualized therapy and disease-specific intervention [35,36]. Moreover, *PEBP1* is significantly associated with poor patient prognosis in stage II colon cancer patients [37]. Normal levels of pPEBP1 are associated with better prognosis than are low levels [37].

In short, in clinical practice, the ferroptosis risk score could be used to guide the more effective clinical practice. We also demonstrated the ferroptosis risk score could be utilized for colon cancer patients’ outcomes. Our data demonstrated the interaction between *NOX1* and BRAF mutation plays an important role in colon cancer. Some limitations of this study should be noted. First, although the prognostic model based on ferroptosis-related genes was validated by an independent validation set, the prognostic ability in other ethnic groups remains unclear. Second, the present study is a bioinformatic analysis, and the potential functional mechanisms between *NOX1* and BRAF mutation were not studied. Hence, further cell and animal studies should be performed to clearly elucidate the relationship between *NOX1* and BRAF mutation.

## 4. Materials and Methods

### 4.1. Colon Cancer Dataset and Preprocessing

The workflow of our study is shown in Appendix A. Public gene-expression data and full clinical annotation were searched for in the GEO and TCGA databases. Patients without survival information were removed from further evaluation. For microarray data from Affymetrix^®^, we applied the “limma” package in R for the analysis. As to datasets in TCGA, RNA sequencing data (FPKM value) of gene expression were downloaded from the Genomic Data Commons (GDC, https://portal.gdc.cancer.gov/ (accessed on 8 June 2020)). The somatic mutation data were acquired from the TCGA database. Data were analyzed with the R (version 3.6.2, 2019-12-12) software and R Bioconductor packages.

### 4.2. Unsupervised Clustering for 15 Prognostic Ferroptosis-Related Genes

The 15 prognostic ferroptosis-related genes included *ABCC1*, *ACACA*, *CISD1*, *CRYAB*, *CS*, *FANCD2*, *FDFT1*, *GLS2*, *GSS*, *HMGCR*, *KEAP1*, *LPCAT3*, *NFS1*, *NOX1*, and *PEBP1* (Figure 1A). Unsupervised clustering analysis was applied to identify distinct vitiligo modification patterns based on the expression of 15 ferroptosis-related genes and classify patients for further analysis. We used the Consensus Cluster Plus package to perform the above steps and 1000 repetitions were conducted to guarantee the stability of the classification.

### 4.3. Immune Profiles in Colon Molecular Subtypes

To quantify the proportions of immune cells in the colon cancer samples, we used the CIBERSORT algorithm, which allows for sensitive and specific discrimination of 22 human immune cell phenotypes. CIBERSORT is a deconvolution algorithm that uses a set of reference gene-expression values (a signature with 547 genes) considered a minimal representation for each cell type and based on those values infers cell-type proportions in data from bulk tumor samples with mixed cell types using support vector regression. The expression value of checkpoint genes is compared between different colon cancer subtypes. Wilcox test was used to conduct different comparisons of the two groups.

### 4.4. Identification of Differentially Expressed Genes (DEGs) between Two Molecular Subgroups of Colon Cancer

DEGs between cluster1 and cluster2 were determined using the R package limma, which implements an empirical Bayesian approach to estimate gene-expression changes using the moderated t-test. DEGs among colon subtypes were determined by significance criteria (adjusted *p* value < 0.05) as implemented in the R package limma. The adjusted *p*-value for multiple testing was calculated using the Benjamini–Hochberg correction.

### 4.5. Functional Enrichment Analysis

Gene annotation enrichment analysis was conducted using the Database for Annotation, Visualization, and Integrated Discovery (DAVID, Frederick, MD, USA, https://david.ncifcrf.gov/ (accessed on 11 April 2021)), an analysis tool for extracting meaningful biological information from multiple gene and protein collections [38]. Up- and down-regulated gene were analyzed separately, a *p* value < 0.05 was considered the threshold value.

### 4.6. Verification of the Ferroptosis Risk Score

The samples in the training and test sets were grouped according to the median value of the ferroptosis risk score in the training set, and then Kaplan–Meier survival curves with log-rank tests were plotted. The following other verifications on the training and test sets were performed at the same time to verify the validity of the ferroptosis risk score: (1) survival status of patients after 1, 3, and 5 years were predicted, and the receiver operating characteristic (ROC) curves were plotted to calculate the AUC. (2) Wilcox and Kruskal test were used to examine the correlation between ferroptosis risk score and clinical information.

### 4.7. Statistical Analysis

With the median value as the cutoff, patients were divided into low or high ferroptosis risk score groups. The survival curves for the prognostic analysis were generated via the Kaplan–Meier method to identify the significance of differences. The forest plot R package was employed to visualize the results of the multivariate prognostic analysis for 60 ferroptosis-related genes in the GEO database. All statistical *p* values were two-sided, with *p* < 0.05 as statistically significant. All data processing was done in R 3.6.2 software (12 December 2019).

## 5. Conclusions

This study demonstrates that the ferroptosis-related genes can be used to classify colon cancer patients based on different clinical and molecular features. A ferroptosis risk score based on the four genes associated with ferroptosis is presented that can independently predict the prognosis of colon cancer patients. Moreover, this study also found the relationship between *NOX1* and BRAF. Given that our results are based on RNA-seq technology, further research is needed to explore the prognostic value of the proposed four gene markers.

## Figures and Tables

**Figure 1 ijms-22-05652-f001:**
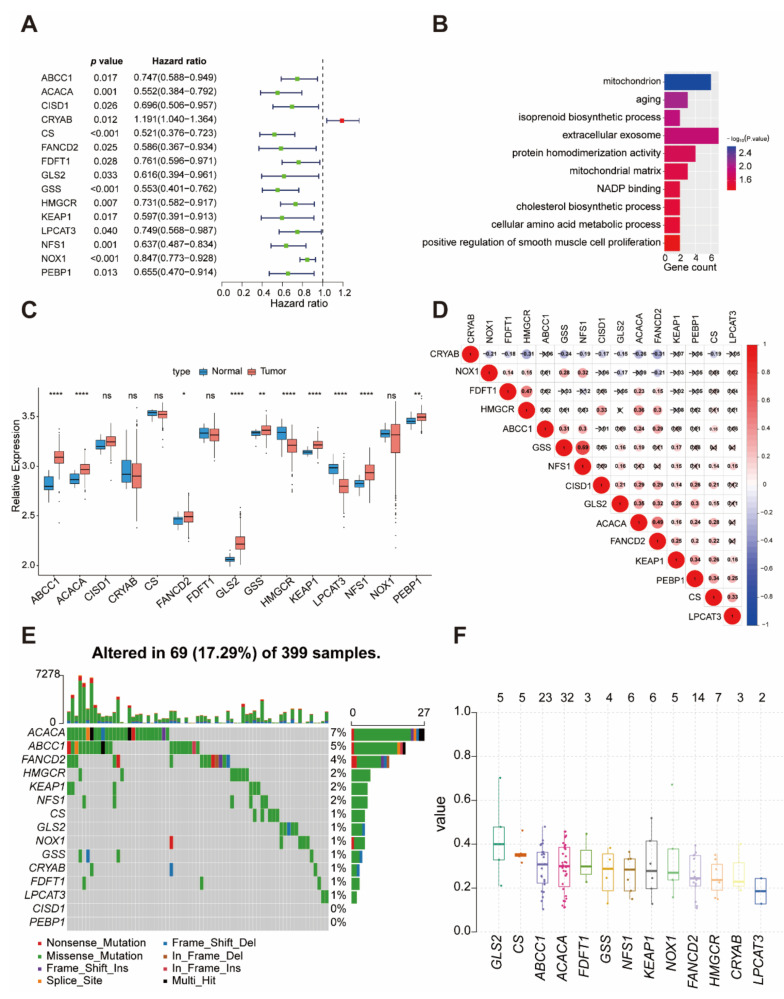
Landscape of genetic and expression variations of prognostic ferroptosis-related genes in colon cancer. (**A**) The prognostic analysis for ferroptosis-related genes in colon cancer using a univariable Cox regression model. (**B**) Functional annotation for 15 prognostic ferroptosis-related genes using GO enrichment analysis. (**C**) The expression of 15 ferroptosis-related genes between normal tissues and colon cancer tissues. Tumor, red; Normal, blue. The upper and lower ends of the boxes represent the interquartile range of values. The lines in the boxes represent median values, and the black dots show outliers. The asterisks represent the statistical *p* value (* *p* < 0.05, ** *p* < 0.01, **** *p* < 0.0001, ns means no significance). (**D**) The interaction between 15 ferroptosis-related genes in colon cancer. (**E**) The mutation frequency of 15 ferroptosis-related genes in 399 patients with gastric cancer from the TCGA-COAD cohort. Each column represents individual patients. The upper bar plot shows TMB. The number on the right indicates the mutation frequency in each regulator. The right bar plot shows the proportion of each variant type. The stacked bar plot below shows the fraction of conversions in each sample. (**F**) Variant allele frequency of 15 ferroptosis-related genes.

**Figure 2 ijms-22-05652-f002:**
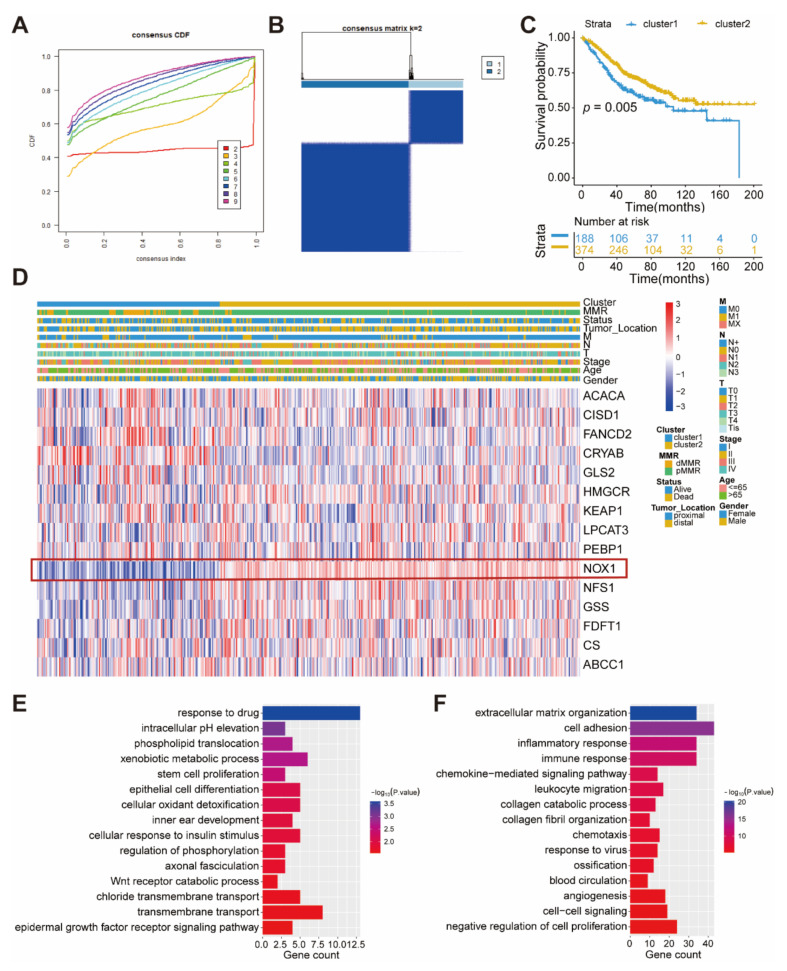
The characteristics of two molecular subgroups of colon cancer. (**A**) The cumulative function (CDF) curves in consensus cluster analysis. CDF curves of consensus score by different subtype numbers (k = 2–9) are represented. (**B**) The consensus matrices of the GSE39582 for k = 2. (**C**) Survival analyses for two molecular subgroups of colon cancer based on 562 patients with colon cancer from the GEO cohorts (GSE39582) including 188 cases in cluster1 and 374 cases in cluster2. Kaplan–Meier curves with Log-rank *p* value 0.005 showed a significant survival difference within two molecular subgroups of colon cancer. (**D**) Unsupervised clustering of 15 ferroptosis-related genes in the GSE39582. The cluster, stage, status, M, N, T, age, and gender were used as patient annotation. Red represents high expression of regulators and blue represents low expression. (**E**) Functional annotation for up expression in cluster2 using GO enrichment analysis. (**F**) Functional annotation for down expression in cluster2 using GO enrichment analysis.

**Figure 3 ijms-22-05652-f003:**
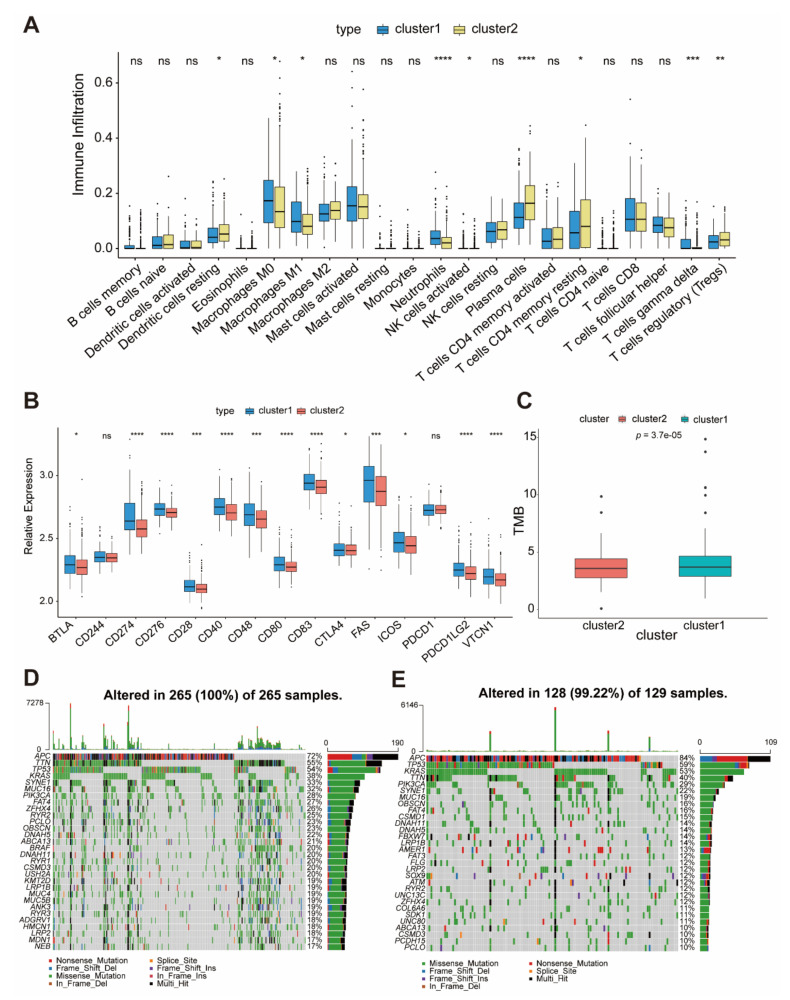
The characteristics of two molecular subgroups of colon cancer. (**A**) The abundance of each immune cell in two molecular subgroups of colon cancer. The upper and lower ends of the boxes represent the interquartile range of values. The lines in the boxes represent median value, and the black dots show outliers. The asterisks represent the statistical *p* value (* *p* < 0.05, ** *p* < 0.01, *** *p* < 0.001, **** *p* < 0.0001). (**B**) The expression of checkpoint genes in two clusters. The upper and lower ends of the boxes represent the interquartile range of values. The lines in the boxes represent median value, and the black dots show outliers. The asterisks represent the statistical *p* value (* *p* < 0.05, *** *p* < 0.001, **** *p* < 0.0001, ns means no significance). (**C**) Difference in TMB between cluster1 and cluster2. (**D**,**E**) The waterfall plot of tumor somatic mutation established by those with cluster1 (**D**) and those with cluster2 (**E**). Each column represents individual patients. The upper bar plot shows TMB. The number on the right indicates the mutation frequency in each gene. The right bar plot shows the proportion of each variant type.

**Figure 4 ijms-22-05652-f004:**
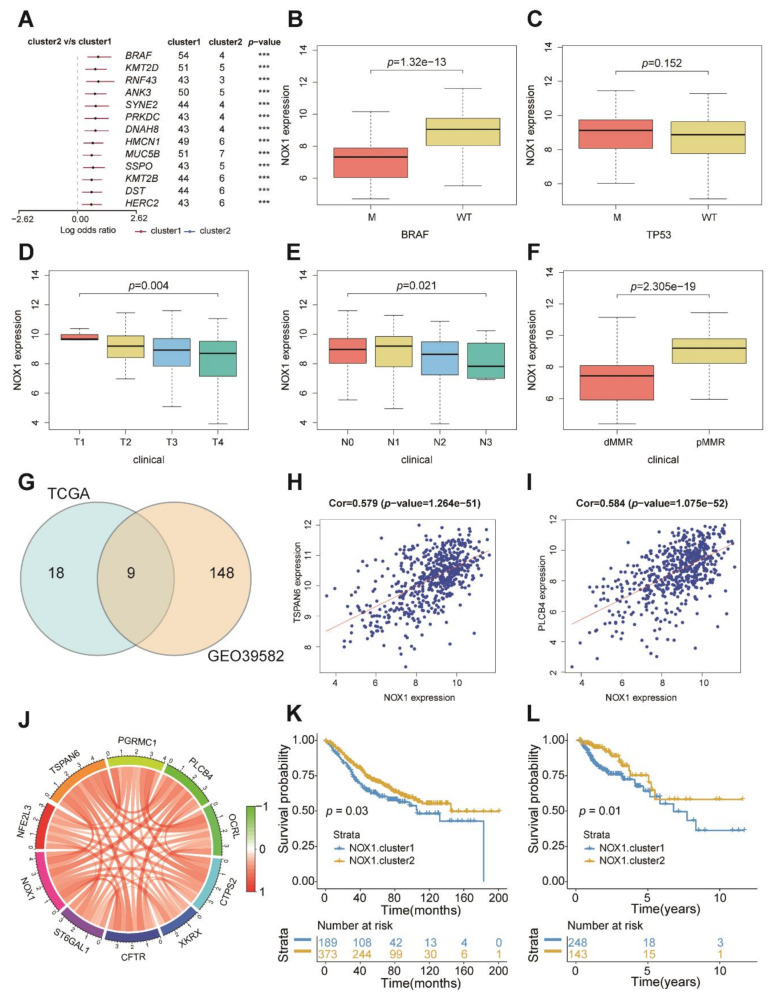
The role of *NOX1* in survival. (**A**) Different mutation genes between cluster1 and cluster2 (**** *p* < 0.0001, ns means no significance). (**B**–**F**) The relationship between *NOX1* expression and clinical features of patients. (**G**) 9 genes related to *NOX1* from the TCGA and GEO39582 databases in Venn diagram. (**H**) The relationship between TSPAN6 and NOX1. (**I**) The relationship between PLCB4 and NOX1. (**J**) The relationship between 9 genes related to NOX1 and NOX1 (**K**,**L**) Survival analyses for patients based on NOX1-related gene expression in GSE39582 (**K**) and TCGA (**L**).

**Figure 5 ijms-22-05652-f005:**
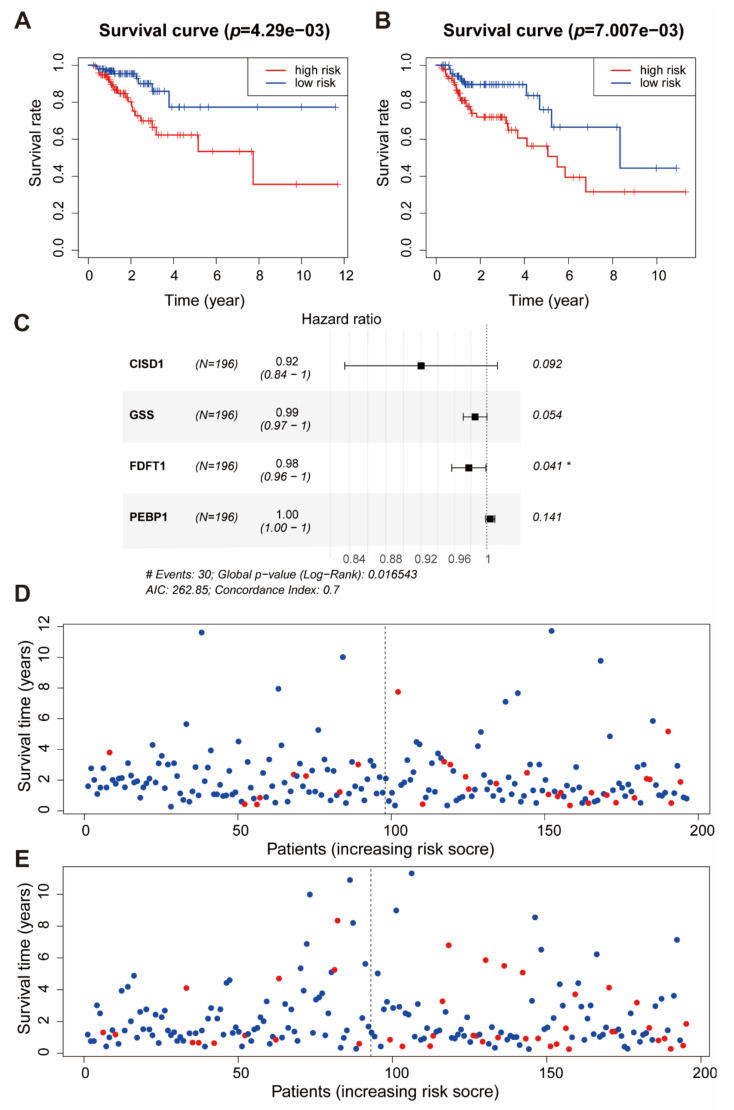
Cox regression model results in TCGA. (**A**) The survival curves of the ferroptosis risk score in the training set. Grouping was based on the median ferroptosis risk score in the training set. Red is the high-level group, and blue is the low-level group. (**B**) The survival curves of the ferroptosis risk score in the test set. Red is the high-level group, and [8] blue is the low-level group. Grouping was based on the median ferroptosis risk score in the training set. (**C**) A forest plot of the multivariate Cox regression model. Hazard ratio is provided in the figure (* *p* < 0.05). (**D**) Patients’ survival status in the training set. The x-axis is the patient ranking in ascending order by the ferroptosis risk score; the y-axis is the survival time. The red dots represent the patients who died, and the green dots represent the surviving patients. (**E**) Patients’ survival status in the test set. The x-axis is the patient ranking in ascending order by the ferroptosis risk score; the y-axis is the survival time. The red dots represent the patients who died, and the green dots represent the surviving patients.

**Figure 6 ijms-22-05652-f006:**
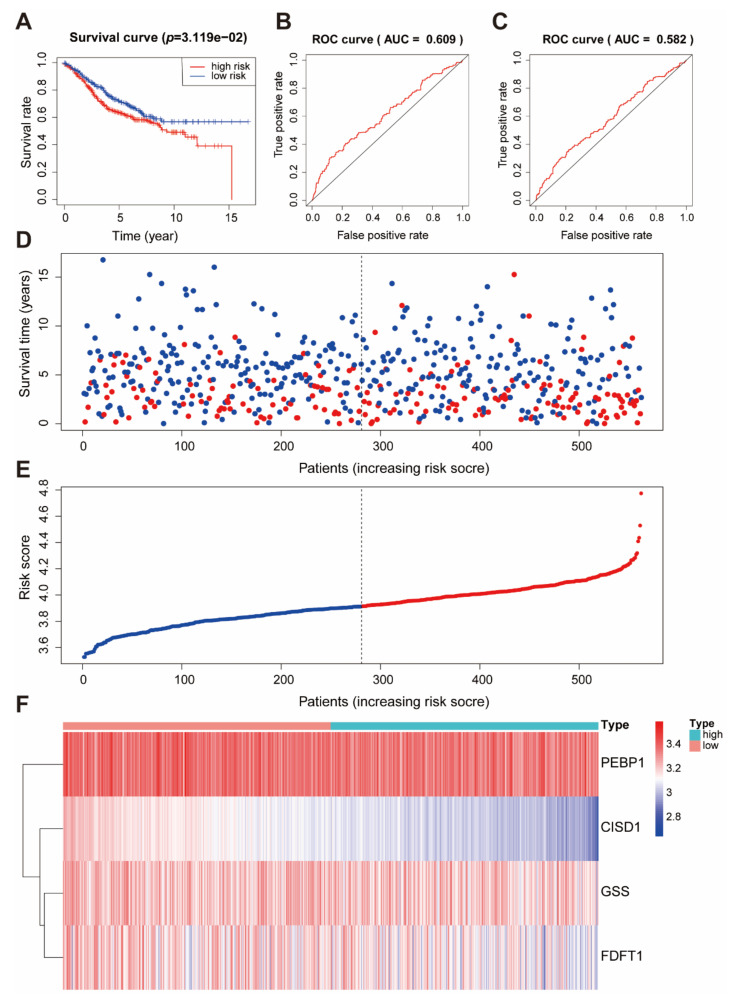
Verification of the prognostic performance of ferroptosis risk score in the GSE39582 cohort. (**A**) The survival curves of the ferroptosis risk score in the GSE39582 cohort. (**B**,**C**) ROC curve of the ferroptosis risk score forecast result after 3 and 5 years. (**D**) Patients’ survival status in the training set. The x-axis is the patient ranking in ascending order by the ferroptosis risk score; the y-axis is the survival time. The red dots represent patients who died, and the green dots represent the surviving patients. (**E**) The distribution and median value of the risk score in the GSE39582 cohort. (**F**) The *PEBP1*, *CISD1*, *GSS*, and *FDFT1* expression in high or low risk score groups.

**Table 1 ijms-22-05652-t001:** Clinical characteristics of patients with colon cancer.

Characteristics	TCGA Cohort		GSE39582 (*n* = 531)
	Training Set (*n* = 196)	Test Set (*n* = 195)	
Age (mean)	67.05	65.77	66.77
Gender (%)			
female	84	94	244
male	112	101	287
Stage			
I	41	25	31
II	71	80	251
III	49	63	190
IV	26	25	59
unknown	9	2	0
T			
T1	8	1	11
T2	40	27	43
T3	121	145	360
T4	27	21	117
Tis	0	1	0
N			
N0	122	108	294
N1	41	53	133
N2	33	34	104

## Data Availability

Publicly available datasets were analyzed in this study. This data can be found here: https://www.cancer.gov/about-nci/organization/ccg/research/structural-genomics/tcga and https://www.ncbi.nlm.nih.gov/gds/ (accessed on 19 May 2021).

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
