# Peer review of "Expression and Prognostic Characteristics of Ferroptosis-Related Genes in Colon Cancer"

_ijms, 2021, doi:10.3390/ijms22115652_

Round 1

Reviewer 1 Report

In a report by Zhu et al, the Authors aims to investigate the prognostic role of NOX1 in colon cancer. Detailed and well performed bioinformatic analysis has been provided. Data are clearly presented. Although the report is performed only in silico, which has been also mentioned by the Authors in conclusions, the quality of data is sufficient to be used by others to create hypotheses and validate experimentally these observations. Importantly, the study touches a relatively novel pathway of cell death (ferroptosis). 

  1. English requires a thorough correction e.g., title: shouldn't be geneS?, line 15 importance--> important etc.
  2. Gene names should be written in italics.

Author Response

Question1、English requires a thorough correction e.g., title: shouldn't be geneS?, line 15 importance--> important etc.

Reply: We have revised the wrong expression, such as ferroptosis related gene--> ferroptosis related genes, line 15 importance-->important.

Question2、Gene names should be written in italics.

Reply:We have described the gene names in italics.

Thanks for your advising!

Reviewer 2 Report

Major point:

Cyclooxygenase-2 (COX-2) is involved in both the inflammatory process and the development of colorectal cancer. It is important that authors take note of the following review (Echizen et al. 2018 Adv Biol Regul DOI: 10.1016/j.jbior.2018.02.001) where NOX1/TNF∝/COX-2 relationships are discussed.
In their work, Zhu et al. need to address NOX1/COX-2 relationships and then link to BRAF.

Author Response

Question: Cyclooxygenase-2 (COX-2) is involved in both the inflammatory process and the development of colorectal cancer. It is important that authors take note of the following review (Echizen et al. 2018 Adv Biol Regul DOI: 10.1016/j.jbior.2018.02.001) where NOX1/TNFα/COX-2 relationships are discussed.
In their work, Zhu et al. need to address NOX1/COX-2 relationships and then link to BRAF.

Reply: We have read the literature carefully, and we have analyzed the relationship between NOX1 and COX2. We found the expression of NOX1 is negative related to COX2. Moreover, In BRAF wild group, the expression of NOX1 is high, while the expression of COX2 is low (Figure S4J-L). Previous studies have showed that COX2 inhibitor, such as nonsteroidal anti-inflammatory drugs (NSAIDs) may decrease the risk and improve prognosis of carcinogenesis of various types of cancer including colorectal cancer. The above results showed that NOX1 genes played a non-negligible regulation role in survival.

Round 2

Reviewer 2 Report

Thank you for taking my comment into account and I am fully satisfied with your response.